# Development of PE/PCL Bilayer Films Modified with Casein and Aluminum Oxide

**DOI:** 10.3390/molecules26113090

**Published:** 2021-05-21

**Authors:** Anita Ptiček Siročić, Ana Rešček, Zvonimir Katančić, Zlata Hrnjak-Murgić

**Affiliations:** 1Faculty of Geotechnical Engineering, University of Zagreb, Hallerova Aleja 7, 42000 Varaždin, Croatia; 2Faculty of Chemical Engineering of Technology, University of Zagreb, Marulićev Trg 19, P.O. Box 177, HR-10000 Zagreb, Croatia; anarescek@gmail.com (A.R.); katancic@fkit.hr (Z.K.); zhrnjak@fkit.hr (Z.H.-M.)

**Keywords:** polyethylene, polycaprolactone, casein, aluminum oxide, barrier properties, water vapor permeability

## Abstract

The studied samples were prepared from polyethylene (PE) polymer which was coated with modified polycaprolactone (PCL) film in order to obtain bilayer films. Thin PCL film was modified with casein/aluminum oxide compound to enhance vapor permeability as well as mechanical and thermal properties of PE/PCL films. Casein/aluminum oxide modifiers were used in order to achieve some functional properties of polymer film that can be used in various applications, e.g., reduction of water vapor permeability (WVTR) and good mechanical and thermal properties. Significant improvement was observed in mechanical properties, especially in tensile strength as well as in water vapor values. Samples prepared with aluminum oxide particles indicated significantly lower values up to 60%, and samples that were prepared with casein and 5% Al_2_O_3_ showed the lowest WVTR value.

## 1. Introduction

Polymers in the industry have a very important place among materials due to their suitability for use in wide applications, especially in relation to their barrier and mechanical properties. Generally, such material must be inert, non-toxic, and strong enough to withstand mechanical forces, i.e., it should act as a barrier system to reduce migration of contaminants. Inadequate barrier properties are critical issues in polymer materials, thus, by incorporating substances with specific properties, better resistance to water vapor and gasses permeability can be achieved. Novel polymer materials based on nanotechnology can provide solutions to increase the polymers performances [1,2,3,4,5]. In many applications of combination of properties, the most effective is use of laminated structures, e.g., bilayer materials where the active particles can be incorporated without degradation of the polymer structure of the carrier film.

Preparation of polymer materials with incorporated nanoparticles includes several procedures such as incorporation during extrusion processes and/or blending in the melt. The other procedures involve coatings with nanoparticles applied onto polymer surfaces, and the final material is then composed of two layers. The main problem due to preparation process is nanoparticles aggregation that can negatively affect the final properties of the final polymer film, e.g., it can cause physical defects in the material. The oxides AlOx are commonly used as barrier coating materials, such as laminated heat-sealable packages for snack foods. Thin Al_2_O_3_ coatings can improve the properties of biopolymers such as casein, enabling the use of these renewable polymers in the production of high-performance materials and pharmaceutical applications. Al_2_O_3_ is non-toxic and non-flammable and has a melting point of 2050 °C [6]. In controlled conditions, Al_2_O_3_ forms highly even and uniform surface films and is, for this reason, considered to be an excellent diffusion barrier.

Casein is a natural biopolymer, and today, biopolymers are used for preparation and characterization of various kinds of biodegradable polymer materials showing properties suitable for a wide range of applications. Casein and casein films are neither too hard nor too brittle, which makes them suitable for the production of thin layers in material, which provide a barrier to gases and water vapor. Although casein films are more permeable to WVTR than the foils, they are able to slow down the transfer of moisture to some extent. As it is known from literature data, a new type of polymer materials made from the milk protein casein is sustainable, biodegradable, safe, and even edible [5,6,7,8,9]. Natural polymers are rarely used in their original form but for certain application must be modified or amended with various additives or adjuvants, such as fillers, pigment, stabilizers, and plasticizers. Its porous structure is suitable as a carrier of nanoparticles, e.g., nanoparticles of aluminum oxide can penetrate in its structure and form useful polymer material.

Low-density polyethylene (LDPE) is chosen due to many of its advantages, particularly: good processability, good mechanical properties, high tensile strength and toughness, flexibility, light-weightedness, water resistance, and great stability. However, its main disadvantage is low oxygen barrier properties. PCL is a versatile synthetic biodegradable polymer with a low melting point, which allows easy processing, and can be blended easily with various amorphous and crystalline polymers to improve the final properties. At room temperature, it has amorphous structure and is soft and rubbery but easily crystallizes and turns into a crystal structure similar to that of PE, with exceptional mechanical properties. One of the main commercial applications of PCL is in the manufacture of biodegradable films and in different biomedical applications [5].

Double-layered polymer films raise an increased environmental concern because of the problems that appear during recycling. There is a growing interest in the development of biodegradable polymers as coating films because it is easy to separate from the surface with chemical or enzymatic methods [5,6,7,8,9]. In such biocoated films, it is necessary to study and test the influence on the properties such as barrier, thermal, and mechanical. Thus, the study of double-layered PE films coated with modified polycaprolactone (PCL) using casein and aluminum oxide (Al_2_O_3_) particles seems very promising.

The main goal was to modify the double-layer polyethylene/polycaprolactone (PE/PCL) films with casein and aluminum-oxide to investigate their effect on barrier, mechanical, and thermal properties.

## 2. Results and Discussion

### 2.1. Water Vapor Permeability

As it is known from the literature, polymeric materials filled with impermeable particles reveal a systematic decrease in permeability [10,11,12]. The results on Figure 1 show that the highest value of water vapor permeability showed a sample of pure polyethylene (34.2 g/m^2^day). The PE/PCL double layer film had a slightly lower WVTR in comparison to the pure PE. It can be seen that sample PE/PCL-Al 1 showed the lowest WVTR value (13 g/m^2^day) in comparison to samples with higher ratio of added aluminum oxide. However, the samples that were prepared with casein/aluminum oxide particles indicated significantly lower values up to 60%. Thus, the sample PE/PCL-C-Al 5 showed the lowest WVTR value (11.2 g/m^2^day). It is assumed that the reason for such behavior may be the partly covalent and the partly ionic characteristics of aluminum oxide. Water molecules have important roles in many processes taking place on the Al_2_O_3_ surface. It is still difficult to identify if water is adsorbed molecularly or dissociatively. It was found that the dissociative adsorption of water is a common mechanism. The adsorption of water on Al_2_O_3_ can be explained by the premise that H_2_O molecules are dissociatively adsorbed on aluminum oxide at all temperatures in the form of OH^.^ and H^.^ radicals [5,13]. The adsorption of the OH^.^ radical took place on the Al^+^ site on an aluminum oxide surface. The unpaired electron on the OH^.^ radical combined with a free electron associated with the Al^+^ site, which resulted in a strong nonpolar bond. Such a mechanism of water adsorption could have been the reason that the samples containing Al_2_O_3_ showed lower water vapor permeability. 

It is known from the literature [14] that the polar water molecules obviously adsorbed more strongly to the surface of Al_2_O_3_, as might be expected given the intrinsically hydrophilic nature of Al_2_O_3_ surface. Generally, samples prepared with casein/aluminum oxide particles indicated a significantly lower WVTR value in comparison with pure PE and PE/PCL films, indicating the good distribution of Al_2_O_3_ in the coating film. Furthermore, biopolymers such as casein have functional surface groups, which can improve the adhesion between the polymer and the Al_2_O_3_ layer. It can be assumed that Al_2_O_3_ particles are homogeneously dispersed in casein because casein has evenly absorbed particles in its micellar structures. Al_2_O_3_ particles were used as the active substance for reducing the permeability of water vapor in the PE/PCL bilayer films, because the particles can disperse homogeneously and can create a polymer structure, which can improve the functional properties of polymer materials [15,16,17,18,19,20,21]. The results of water vapor permeability, achieved in two samples (PE/PCL-C-Al 5 and PE/PCL-C-Al 10), were close to the required level (0.1–10 g/m^2^day) for commercial dry food packaging [15]. The WVTR of polymeric materials usually decreases with increased coating layer of film thickness if cracking of the barrier layer does not occur. As a slight increase of WVTR value was obtained for sample PE/PCL-C-Al 10, it can be concluded that, in this sample, some defects appeared as cracks, voids, and pinholes, and/or permeation could have taken place through the amorphous regions of polymer films. A reduced value of water vapor permeability is important for polymer materials, especially for packaging materials, because this kind of packaging provides prolongation of the freshness of products [15].

### 2.2. TG Analysis

The results of TG analysis of PE/PCL bilayer films are shown in Table 1 and Figure 2. The values of bilayer films showed a higher thermal stability compared to pure PE film, as can be seen from T_95_ and T_max_ values, although the maximum speed of degradation (v_max_) of pure PE was much lower due to different kinetics of thermal decomposition. As it is known from the literature [22,23,24], initial degradation of polyethylene starts at 398 °C, thus it can be concluded that addition of aluminum oxide and casein/aluminum oxide delays the decomposition process. Significantly higher thermal stability was shown in PE/PCL in comparison to pure PE film, as the initial decomposition temperature (T_95_) as well as the T_max_ were higher for 10 °C, i.e., all other measured parameters pointed to better stability. Additionally, the temperature of maximum degradation rate (T_max2_) for all samples was in the range from 465 °C to 478 °C, while the values of initial temperatures (T_95_) showed even higher differences. PE/PCL-Al 1, PE/PCL-Al 5, and PE/PCL-Al 10 samples had a slightly higher initial decomposition temperature in comparison to samples prepared with casein/aluminum oxide. Increase of T_95_ and T_max_ values caused enhancement of the maximum degradation rate, i.e., from 19%/min for pure polyethylene to 29.1%/min for PE/PCL-Al 5. It can be assumed that aluminum oxide particles established an interaction with PCL polymer, i.e., the mechanism and the kinetics of thermal decomposition of polymers were modified. The altered mechanism of thermal decomposition due to different sample composition was also indicated by the appearance of broad maximum in the temperature range 385–425 °C, shown in Table 1 denoted as T_max1_. Samples containing Al_2_O_3_ in low concentrations of 1 wt% showed a similar decomposition mechanism as pure PE. The best thermal stability showed PE/PCL-Al 5 and PE/PCL-Al 10 with the highest values of T_max2_ at 477 °C, which was about 12 °C higher than pure PE. Samples prepared with casein/aluminum oxide showed also good thermal stability, slightly lower than samples prepared with aluminum oxide. It was obvious that thermal decomposition began at higher temperatures, and the degradation rate was accelerated with the addition of aluminum oxide as well as casein/aluminum oxide. As the rate of thermal decomposition increased, it was obvious that the mechanism of decomposition changed. Many studies reported that metal ions in oxide usually act as catalysts for the degradation of hydroperoxides, which increases decomposition of the polymer material due to primary chain scission. Presence of free radicals in the beginning of the thermal decomposition can affect the degradation mechanism and the decomposition process [25,26,27,28,29]. From the point of thermal stability, mechanisms, and kinetics of thermal decomposition, it can be concluded that samples prepared with 5 and 10 wt% of Al_2_O_3_ with and without casein had similar thermal behavior as the PE/PCL sample, while the samples with low concentrations of Al_2_O_3_ had similar behavior to those of pure PE.

### 2.3. Mechanical Properties

The mechanical properties of the investigated PE/PCL films are presented in Figure 3 and Figure 4. It is obvious that values significantly increased. On the other hand, values for neat PE/PCL film in comparison to pure PE indicated some deterioration of properties. The elongation at break of PE/PCL samples was decreased in respect to neat polymers (PE and PCL) because the samples were prepared as the two-layer film of different thicknesses, not as the polymer blends. As it is known from the literature [25,30,31], PCL is a semirigid polymer with high elongation at break (700%) and low tensile strength (23 MPa) and, as such, causes deterioration of mechanical properties. Addition of Al_2_O_3_ significantly increased the elongation at break and the tensile strength of PE/PCL samples compared to pure PE polymer. For example, PE/PCL-AL 5 showed higher stretching of 36% and tensile strength of 43%. A good interaction of aluminum oxide with PCL polymer led to a large interfacial area between matrix and modifier, which affected changes in molecular mobility, resulting in better mechanical properties. Many studies showed [24,25,32,33,34] that, when dispersion of inorganic particles in a polymer matrix was improved due to established interactions, the stress was more effectively transferred from the polymer to the filler, and, thus, properties were enhanced.

Nevertheless, the mechanical properties of PE/PCL samples prepared with casein/aluminum oxide significantly decreased with the increase of Al_2_O_3_ ratio. The lowest values of elongation at break and tensile strength were shown in the PE/PCL-C-Al 10 sample, indicating poor interaction between the components, which could result in inhomogeneous distribution of particles and/or presence of agglomerates in a polymer matrix. The interphase between a polymer and a filler often acts as a “weak point” in the polymer material and causes premature breakage. Therefore, it is important to ensure good dispersion of each component in the polymer matrix and prevent the formation of agglomerates that markedly weaken the mechanical properties [34].

### 2.4. FTIR Analysis

Studied PE/PCL films were recorded on the polyethylene side and on the PCL coating side, and results are shown in Figure 5a,b. On spectrogram 5a are visible characteristic vibrational bands for pure PE, i.e., aliphatic groups C-H at 2915 cm^−1^ and 2848 cm^−1^ and at 1471 cm^−1^ corresponding to -CH_3_ stretching, while a peak at 718 cm^−1^ corresponded to -CH_2_ group in the vinyl group. Typical vibrational bands for PCL polymer are shown in the spectrogram 5b. Absorption of vibrations at 2915 cm^−1^ and 2848 cm^−1^ corresponded to C-H groups for asymmetric and symmetric stretching, respectively. The peak at 1725 cm^−1^ corresponded to carbonyl groups C=O and asymmetric stretching O-C-O groups at 1244 cm^−1^, and their symmetric stretching showed a peak at 1164 cm^−1^ as well as a peak at 1191 cm^−1^, showing the visible O-C-O bond on the vibrational bands [35,36]. The peak at 2848 cm^−1^ showed significantly lower absorbance intensity, which was due to the increase in hydrogen bonding [37,38]. FTIR spectra recorded for the modified PE/PCL samples did not show notable changes because casein was present in very low concentrations, and vibrational bands of Al_2_O_3_ were below 500 cm^−1^ [39]. Higher concentration of Al_2_O_3_ resulted in lower absorbance intensity of PE and PCL polymer vibrational bands. 

### 2.5. SEM Analysis

Analysis of the structure of prepared PE/PCL films is presented in Figure 6. As we can see in Figure 6a, morphology of the PE/PCL-Al-1 was homogenous and uniform with small aggregates; otherwise, most of the particles were randomly and homogenously distributed in the polymer matrix. Micrographs 6b,c show approximately similar morphology with the presence of aggregates at coarse surface of the polymer matrix. As it is known from the literature, dispersion of particles is a critical parameter that has influence on properties of material, because some aggregation may cause fracture of material or an increase in the permeability of low molecules such as water [15,24,25]. Morphology of PE/PCL samples prepared with casein and aluminum oxide are presented in micrographs 6d–f. The PE/PCL sample prepared with casein and Al_2_O_3_ showed better morphology in comparison to other PE/PCL samples. Micrographs 6e,f show slightly coarser morphology as a consequence of the formation of a PCL film from the solution showing the uneven surface of the film. The samples in micrographs 6d–f show lower aggregations, and it was assumed that such behavior was due to the presence of the casein polymer. Casein structure is suitable for modification because Al_2_O_3_ particles can penetrate into its micellar structure, which enables homogeneous distribution in the polymer matrix. The structure of casein micelles significantly depends on temperature, ionic strength between bonds, water activity, and other parameters, which can affect particle size as well as content of sub-micelles in casein. For that reason, preparation conditions are very important and can notably affect the size of aggregates of casein micelles as well as the final morphology of prepared polymer material [5]. Generally, chemical compatibility between the matrix and the filler plays a critical role in the particle dispersion, such as Al_2_O_3_ within the matrix and the adhesion between both phases. Consequently, we can conclude that the presence of casein increases the miscibility between the PCL polymer and Al_2_O_3_, which affects the improvement of properties of the final material.

## 3. Experimental Part

### 3.1. Materials

Commercial PE film (LDPE) (thickness of 60 μm, density 0.92–0.95 g/cm^3^, grammage: 73.6 g/m^2^) was coated with the polycaprolactone (PCL) by Sigma Aldrich (Munich, Germany), (Mn = 70,000–90,000 g/mol, density 1.145 g/cm^3^, boiling point 60 °C). As a solvent, tetrahydrofuran (THF) obtained by LachNer (Neratovice, Czech Republic) was used (Mn = 72.11 g/mol, density of 0.89 g/cm^3^, and boiling point of 65 °C). Casein powder was obtained from Across Organics (Geel, Belgium) (Mn = 477.55 g/mol, protein content ≥ 92%, density 1.25 g/cm^3^, melting point of 280 °C). Particles of aluminum oxide (Al_2_O_3_) were obtained by Sigma Aldrich (Munich, Germany) (particle size ≤ 13 nm, purity ≥ 98%).

### 3.2. Preparation of Casein/Aluminum Oxide Mixture

Casein solution was prepared by dissolving 4 g of sodium chloride in 700 mL of distilled water to give a 0.1 M solution of NaCl. In this solution, 60 g of casein was gradually added. Dissolution of casein was carried out on a magnetic stirrer at elevated temperature (37–38 °C) until complete dissolution. In such prepared casein solutions, Al_2_O_3_ particles were added in various ratios given in Table 2. Samples were left for 24 h at room temperature. The liquid layer above the sedimented solid particles was carefully separated by decantation, and the remaining solution was centrifuged at 3000 rpm for 15 min. The obtained precipitate was washed with distilled water, centrifuged again, and dried in an oven at 35 °C for 6 h. After that, samples were stored in the freezer.

### 3.3. Preparation of Double Layer PE/PCL Films

To obtain a PCL coating layer modified with casein/aluminum oxide mixture, a 10% mass solution of PCL was prepared in THF using a magnetic stirrer at room temperature at a speed of 350 rpm. In prepared PCL solution (50 mL) was added 2 g of casein/aluminum oxide mixture, which was dispersed for 8 min at a speed of 15,000 rpm by IKA T25 digital ULTRA-Turrax disperser (IKA, Staufen, Germany). The result was 10% mass solution of PCL containing 2% Al_2_O_3_ in PE/PCL-C-Al 10, 1% Al_2_O_3_ in PE/PCL-C-Al 5, and 0.2% Al_2_O_3_ in PE/PCL-C-Al 1. On the surface of PE films (14 × 14 cm), the PCL polymer solution was loaded in thickness of 30 µm using the Zehntner ZFR 2040 applicator (Zehntner Testing Instruments, Sissach, Switzerland). The samples were dried (5–7 min), and the final thickness of PE/PCL films was 63 µm.

### 3.4. Characterization

#### 3.4.1. Water Vapor Permeability 

Permeability determination of water vapor of double layer PE/PCL films was carried out on the Herfeld’s appliance. Herfeld’s device consists of a glass container with a metal lid which contains a circular hole diameter of 36 mm. Into the glass vessel was poured 50 cm^3^ of water. The cover of the machine set the pattern of a circular diameter of 55 mm (face up), and the lid was closed. The device was placed in a desiccator with 97% sulfuric acid. The weight of the apparatus with the specimen and the water was determined at the beginning and after the given time intervals of 24 h and 48 h. The results are expressed as the average of triplicate determinations. 

Water vapor transmission (WVTR) was determined according to Equation (1):(1)$$\mathrm{WVTR}=m_{0}-\frac{m_{2}+m_{3}}{2}$$
*m*_0_—mass of the device with water and specimen at the beginning (g)*m*_2_—mass of the device with water and a test tube after 24 h (g)*m*_3_—mass of the device with water and a test tube after 48 h (g)

#### 3.4.2. TG Analysis

Thermogravimetric analysis (TGA) of studied samples was carried out using a TA Instruments Q500 analyzer (TA Instruments, Newcastle, New Jersey). The results were obtained for the temperature range from 25 °C to 550 °C at heating rate of 10 °C/min under the flowing N_2_ atmosphere. Nitrogen was used as a carrier gas with a constant flow rate of 60 mL/min during analysis. The residual amount of material was determined (m_r550 °C_) along with the temperature at which 5 mass percent of the sample was decomposed (T_95_), along with the temperature at the maximum degradation rate (T_max_) and the maximum degradation rate (v_max_).

#### 3.4.3. Mechanical Properties

For the testing of mechanical properties (tensile strength and elongation at break) of PE/PCL bilayer films, a universal mechanical testing machine, the Zwick Testing Machine (Model 1445, Zwick GmbH& Co.KG, Ulm, Germany), was used. Measurements were carried out according to ISO 527-1 and ISO 527-2 standards, and the samples were prepared in dimensions (100 × 10 mm). A gauge length of 25 mm and a crosshead speed of 50 mm/min were used at constant temperature (23 °C) and humidity (50% RH). The results were expressed as the average of five measurements.

#### 3.4.4. FTIR Spectroscopy

Prepared PE/PCL samples were also characterized by attenuated total reflectance Fourier transform infrared spectroscopy (ATR FTIR), Spectrum One FTIR spectrometer, Perkin Elmer (Boston, MA, USA), in the range from 4000 to 650 cm^−1^ with resolution of 4 cm^−1^.

#### 3.4.5. SEM Analysis

Scanning electron microscopy (SEM) was utilized to examine the morphology of the cross section of the PE/PCL samples using Tescan VEGA 3 SEM at 10 kV (Brno, Czech Republic). Samples were cryogenically fractured in liquid nitrogen and then gold sputtered for examination.

## 4. Conclusions

Requirements of polymeric materials that can be used in many applications are good properties that especially include mechanical, thermal, and barrier properties. The main disadvantages of polymer materials are their relatively poor barrier properties. There are two ways of improving them: incorporation of active substances in the polymer film and production of materials with a layered structure. The active substances can slow down oxygen diffusion, carbon dioxide, as well as water vapor. Barrier properties, e.g., reduced permeability to water vapor of PE/PCL films prepared with casein/aluminum oxide, were significantly improved and showed a reduction of values up to 60% in comparison to the pure PE and PE/PCL films in this study. Thermal stability of PE/PCL films was enhanced due to addition of casein and aluminum oxide particles. The addition of aluminum oxide and casein in PE/PCL films significantly affected the improvement of mechanical properties. Thus, the strength of the PE/PCL-Al 5 sample compared with pure PE increased from 12.9 N/mm^2^ to 18.5 N/mm^2^ and stretched from 533% to 727%. The micrographs show that incorporation of aluminum oxide and casein in the PE/PCL samples caused uniform distribution of the particles regardless of the formation of certain agglomerates and cracks. Nowadays, most polymer materials are practically non-degradable, representing a serious global environmental problem, thus there are many efforts to produce materials which improve the quality of products whilst reducing packaging waste.

## Figures and Tables

**Figure 1 molecules-26-03090-f001:**
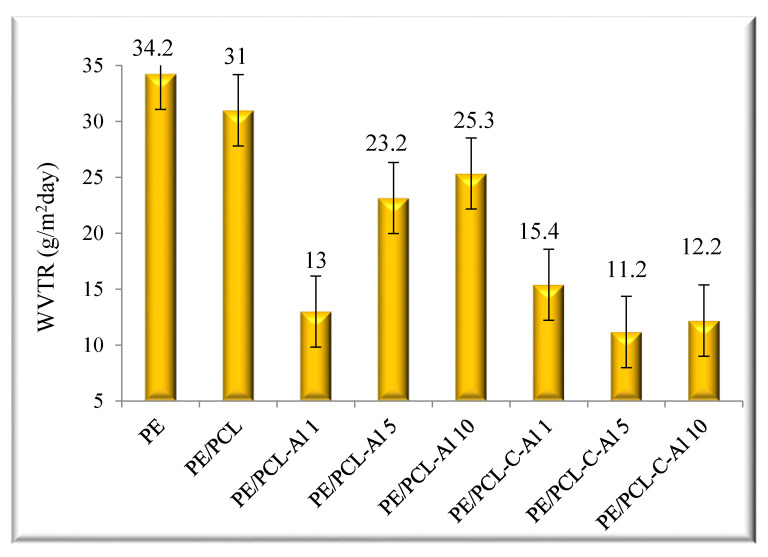
The water vapor permeability of pure PE, PE/PCL, and modified PE/PCL films.

**Figure 2 molecules-26-03090-f002:**
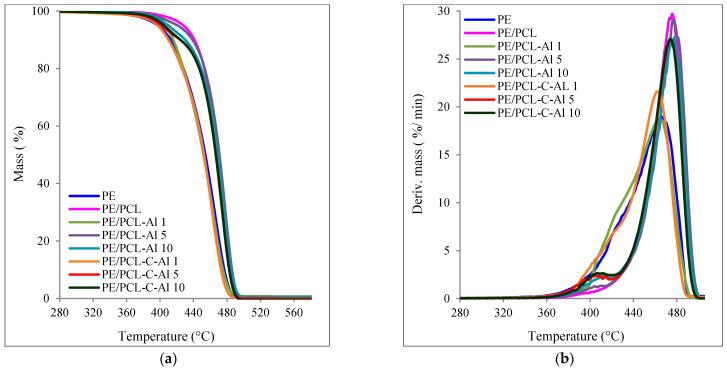
(**a**) TG and (**b**) DTG curves of PE bilayer films.

**Figure 3 molecules-26-03090-f003:**
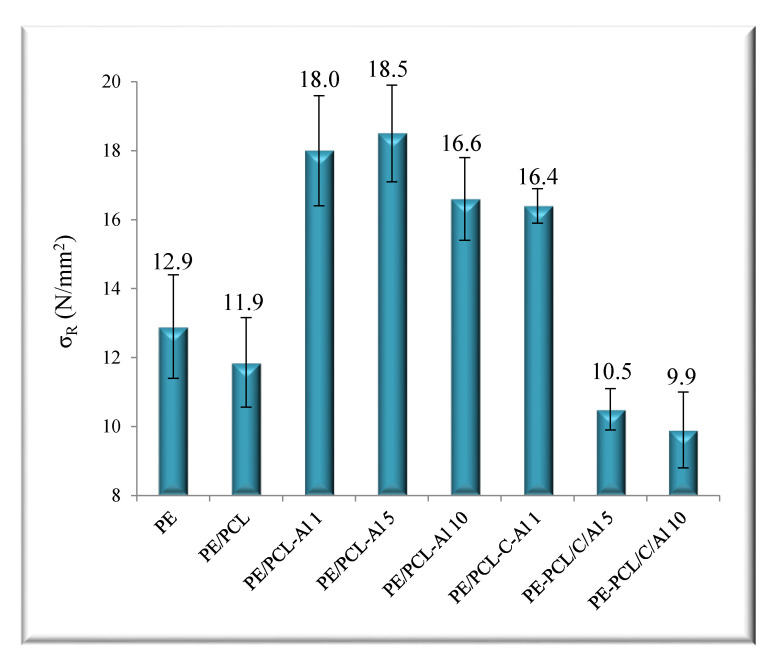
Tensile strength (σ) of PE/PCL bilayer films.

**Figure 4 molecules-26-03090-f004:**
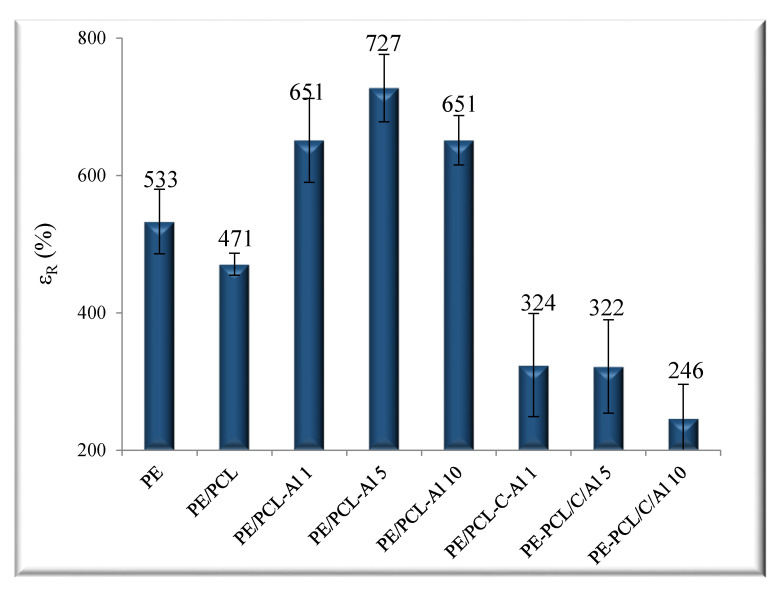
Elongation at break (ε) of PE/PCL bilayer films.

**Figure 5 molecules-26-03090-f005:**
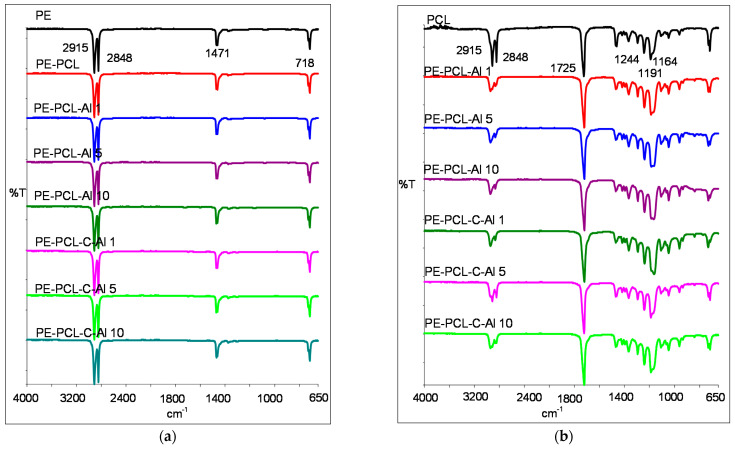
FTIR spectra of PE/PCL films: (**a**) on PE side, (**b**) on the side of the PCL coating.

**Figure 6 molecules-26-03090-f006:**
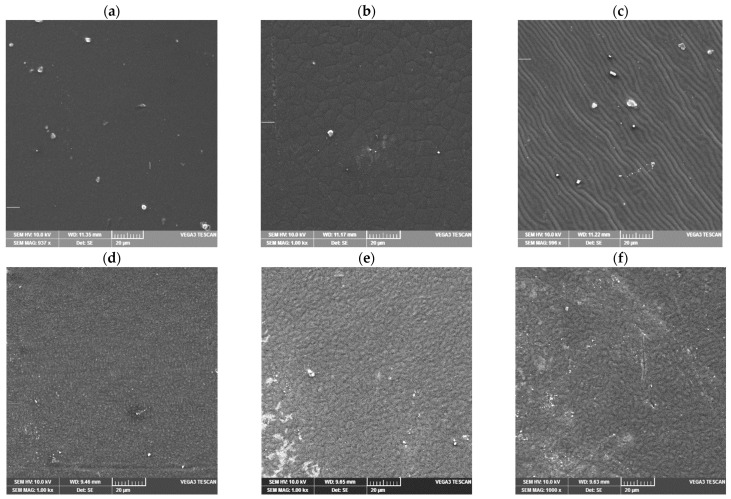
SEM micrographs of PE/PCL films: (**a**) PE/PCL-Al 1, (**b**) PE/PCL-Al 5, (**c**) PE/PCL-Al 10, (**d**) PE/PCL-C-Al 1, (**e**) PE/PCL-C-Al 5, (**f**) PE/PCL-C-Al 10.

**Table 1 molecules-26-03090-t001:** Decomposition beginning temperature (T_95_), the temperature at the maximum degradation rate (T_max_), the char residue (m_r550 °C_), and the maximum degradation rate (v_max_) of PE/PCL bilayer films.

Sample	T_95_ (°C)	T_max1_ (°C)	T_max2_ (°C)	m_r550 °C_ (%)	v_max_ (%/min)
PE	397.1	415–430	465.9	0.06	19.0
PE/PCL	427.9	–	475.1	0.01	29.3
PE/PCL-Al 1	406.5	398–435	464.8	0.72	11.3
PE/PCL-Al 5	421.5	385–425	477.1	0.83	29.1
PE/PCL-Al 10	409.9	385–425	477.9	0.54	27.3
PE/PCL-C-Al 1	397.4	385–427	463.1	0.14	21.6
PE/PCL-C-Al 5	401.0	385–425	475.6	0.01	26.9
PE/PCL-C-Al 10	404.5	385–425	474.8	0.50	27.0

**Table 2 molecules-26-03090-t002:** Samples with casein/aluminum oxide particles, (*w*/*w*)/%.

Sample	C	Al
C-Al 1	99.0	1.0
C-Al 5	95.0	5.0
C-Al 10	90.0	10.0

C—casein, Al—Al_2_O_3_.

## Data Availability

Data available in a publicly accessible repository.

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
