# Peer review of "Development of PE/PCL Bilayer Films Modified with Casein and Aluminum Oxide"

_molecules, 2021, doi:10.3390/molecules26113090_

Round 1

Reviewer 1 Report

The topic regarding the paper “Modified PE/PCL bilayer films as packaging material with casein and aluminium oxide nanoparticles”, is of interest for the scientific community, however this contribution can not be published as it.

The paper requires huge improvements since lack in experimental investigation, in the description of work and in the consistence of results.

Specifically:

  • Authors reports that formulate materials can be used for packaging application, addressing the work in this view. WVTR analysis were carried out, however O2/CO2 permeability is needed to observe benefits in terms of shelf life.
  • Moreover, Contact angle, SEM characterization, transparency index could be provide more information on the formulated systems. For example, morphology characterization is needed to be sure that nanoparticle are well distributed into the matrix, allowing a comprehensive evaluation of all findings, mechanical properties included.
  • The results are very questionable. From one hand the addition of casein and nanoparticles decreased WVTR values, although it is enough to fill just with the 1% of Al2O3 to obtain better results. From the other hand, the same materials (PE/PCL/C Al 5 and PE/PCL/C Al 10) provided the worst outcome in terms of mechanical properties.

For all the reasons reported, I am sorry to reject this paper in this form.

Author Response

Dear reviewer,

we have made some improvements in Manucstript according to the reviewer suggestions. 

Regards 

Reviewer 2 Report

The author used casien and Al2O3 coating into PCL film. The data is fine but the aim of this study is not clear. what is purpose to use casien and Al2O3?

Does anyone use these two casien and Al2O3 in PCL film , Al2O3 is toxic to human and casien protein is nutrient for bacteria growing. I donot think it is good to use these two materials into PCL film. 

although it decrease  WVTR value, and what is big deal for lower WVTR values?

  1. in introduction line 23 to 45, you need to talk about casien and Al2O3, why is so important in bilayer film and why choose these two?
  2.  line 23 to 45, there is no focus and not related to this article.

Author Response

Dear reviewer, please find the answer on your comment in attach.

Regards

Reviewer 3 Report

General comments:

Barrier properties are discussed only in terms of water vapour permeability; however gas (e.g. O2, CO2) permeability, which is very important for packaging materials, is not discussed. Also, when considering final use as packaging material, the effect of material on microbial activity is of very strong significance. What if the benefit of decreasing WVTR is negatively counterbalanced by possible enhanced microbial growth (e.g. due to presence of casein)? At least discussion based on literature data should be made. While the initial amount of Al2O3 added to casein solution is known, there is no information on the Al2O3 amount remaining in the C-Al system after the preparing procedure described in Section 2.2 (part of Al2O3 remains as sedimented solid particles – line 78). Therefore the behavior of samples cannot be correctly correlated according to Al1, Al5, Al10 amount. Even if Al2O3 has initial nanosize, possible agglomeration might occur, leading to dispersion out of nanoscale. Therefore “nano” should be more carefully utilized.

Other comments:

Line 65-67: Check if 60 oC for PCL and 65 oC for THF refer to boiling point and melting point, respectively, or switch should be made.

Line 90: Why the casein/aluminum oxide material is called nanocomposite? It might be a simple mixture.

Table 1: Why not C-Al 0 to see the differences?

Line 118: The material remaining after heating at 550 oC cannot be considered nanocomposite; “yield” is not at its good usage here – “amount” would be better.

Section 3.1, WVTR: While WVTR decreased with addition of Al1 on PE/PCL, it is not explained why WVTR increased with increasing Al5 and Al10. Variation of WVTR with Al content (minimum value for Al5) in PE/PCL-C should be discussed. Why Al2O3 was used to decrease WVTR, since it has hydrophilic nature (line 151-152)? Discussion should be made on this aspect. Good distribution of Al2O3 particle in final films is only speculated, but no proof is presented. While micellar structure (line 156) might be considered in solution, this is strongly disrupted at solvent evaporation.

Section 3.2, TG: Table 2 – why mr550 does not increase with increasing Al2O3 content (see PE/PCL-Al 10 and PE/PCL-C-Al 5)? Tmax1 corresponds, in fact, to shoulder, for which one cannot consider a clear “Tmax”. Sample humidity (mass loss below ~ 110 oC) is not considered; this strongly affects the correctness of T95 and mr550. Behaviour of Al5 and Al10 samples has high degree of similarity, both for PE/PCL and for PE/PCL-C, which is strongly different to Al1 samples; this should be discussed. The temperature for the beginning of thermal degradation strongly depends on experimental parameters therefore the value of 398 oC (line 173), even if supported by literature citations, is only a particular case. One cannot correctly discuss on the delay of thermal degradation since the very initial moments are not observed. Fig. 2 should include the region below 380 oC since at that temperature degradation is already started. T95 should be better replaced by T5 (temperature for 5% mass loss). However, while this parameter is largely used in industry, it does not correctly represent the thermal stability from the point of view of the first moments of the process. The provided TG data cannot support the affirmation of no Al2O3 agglomerates (lines 185-186). In fact, if this would be true, it will be a sign of agglomerates in Al1 samples, which would be very curious.

Section 3.3, Mechanical properties: Why PCL, with much higher elongation at break, decreased the value for PE/PCL?

Section 3.4, IR: Since ATR cannot penetrate the depth of 60 um of the PE film, it is not expected to see any differences among samples with various treatments on the other side (Fig. 5.a). If no notable change is observed among samples on the PCL side, what is the meaning of Fig. 5.b?

Author Response

Dear reviewer,

please find the answer on your comments in attach.

Regards

Round 2

Reviewer 1 Report

The paper was highly improved as requested. I recommend it for publication in this revised version

Author Response

Thank you very much for the comments.

Reviewer 2 Report

1. figure 1 and 3, the error bars need to be consist, put the error bars into figures.

2. put the all films pictures as one of figures.

3. english need to be revised by native english  editor. 

Author Response

Dear reviewer,

Reviewer 3 Report

General comments: Authors made no improvement at fundamental level. Permeability to gases (O2, CO2) and microbial activity was still not addressed; therefore reference to potential use of materials as packaging should be removed, or, at least, it should be mentioned what kind of packaging is considered where only water vapour permeability is important. The real amount of Al2O3 in final samples was still not addressed, therefore discussion based on variation of Al2O3 amount do not have strong support.

Particular comments:

- “composites” should be removed from keywords and from line 226, since there is no proof that the obtained materials have this type of structure.

- “nanoparticles” should be removed from the legend of Table 1 and from everywhere else in text when referring to obtained materials, since no proof was presented that the Al2O3 preserve the nano size and no agglomeration occurred.

- Line 88: replace “boiling point” with “melting point”

- Figure 2: The X scale for TG (2.a) should be also extended to 280 oC, as for DTG (2.b)

Author Response

Dear reviewer,
